# Tunable colloid trajectories in nematic liquid crystals near wavy walls

Yimin Luo [1], Daniel A. Beller [2], Giuseppe Boniello[1], Francesca Serra [3] & Kathleen J. Stebe [1]

The ability to dictate the motion of microscopic objects is an important challenge in fields ranging from materials science to biology. Field-directed assembly drives microparticles along paths defined by energy gradients. Nematic liquid crystals, consisting of rod-like molecules, provide new opportunities in this domain. Deviations of nematic liquid crystal molecules from uniform orientation cost elastic energy, and such deviations can be molded by bounding vessel shape. Here, by placing a wavy wall in a nematic liquid crystal, we impose alternating splay and bend distortions, and define a smoothly varying elastic energy field. A microparticle in this field displays a rich set of behaviors, as this system has multiple stable states, repulsive and attractive loci, and interaction strengths that can be tuned to allow reconfigurable states. Microparticles can transition between defect configurations, move along distinct paths, and select sites for preferred docking. Such tailored landscapes have promise in reconfigurable systems and in microrobotics applications.

[1] Chemical and Biomolecular Engineering, University of Pennsylvania, Philadelphia, PA 19104, USA. [2] University of California, Merced, CA 95343, USA. [3] Physics and Astronomy, Johns Hopkins University, Baltimore, MD 21218, USA. Correspondence and requests for materials should be addressed to F.S. (email: fserra1@jhu.edu) or to K.J.S. (email: kstebe@seas.upenn.edu)

Ever since Brown discovered the motion of inanimate pollen grains, material scientists have been fascinated by the vivid, life-like motion of colloidal particles. Indeed, the study of colloidal interactions has led to the discovery of new physics and has fueled the design of functional materials[1–3]. External applied fields provide important additional degrees of freedom, and allow microparticles to be moved along energy gradients with exquisite control. In this context, nematic liquid crystals (NLCs) provide unique opportunities[4]. Within these fluids, rod-like molecules co-orient, defining the nematic director field[5]. Gradients in the director field are energetically costly; by deliberately imposing such gradients, elastic energy fields can be defined to control colloid motion. Since NLCs are sensitive to the anchoring conditions on bounding surfaces[6,7], reorient in electro-magnetic fields[5,8], have temperature-dependent elastic constants[5] and can be reoriented under illumination using optically active dopants[9,10], such energy landscapes can be imposed and reconfigured by a number of routes.

Geometry, topology, confinement, and surface anchoring provide versatile means to craft elastic energy landscapes and dictate colloid interactions[11–14]. This well-known behavior[4,15] implies that strategies to dictate colloidal physics developed in these systems are robust and broadly applicable to any material with similar surface anchoring and shape. Furthermore, the ability to control the types of topological defects that accompany colloidal particles provides access to significantly different equilibrium states in the same system. Thus, the structure of the colloid and its companion defect dictate the range and form of their interactions.

By tailoring bounding vessel shape and NLC orientation at surfaces, one can define elastic fields to direct colloid assembly[4]. This was shown for NLC controlled by patterned substrates[16,17], optically manipulated in a thin cell[18], or in micropost arrays[19,20], grooves[21–23], and near wavy walls[24,25]. In prior work, the energy fields near wavy walls have been exploited to demonstrate lock-and-key interactions, in which a colloid (the key) was attracted to a particular location (the lock) along the wavy wall to minimize distortion in the nematic director field. However, the elastic energy landscapes obtainable with a wavy wall are far richer, and provide important opportunities to direct colloidal motion that go far beyond near-wall lock-and-key interaction.

In this system, elastic energy gradients are defined in a non-singular director field by the wavelength and amplitude of the wavy structure, allowing long ranged wall-colloid interactions. Colloids can be placed at equilibrium sites far from the wall that can be tuned by varying wall curvature. Unstable loci, embedded in the elastic energy landscape, can repel colloids and drive them along multiple paths. In this work, we develop and exploit aspects of this energy landscape to control colloid motion by designing the appropriate boundary conditions. For example, we exploit metastable equilibria of colloids to induce gentle transformations of the colloids' companion topological defects driven by the elastic fields. Since topological defects are sites for accumulation of nanoparticles and molecules, such transformations will enable manipulation of hierarchical structures. We also create unstable loci to direct particle trajectories and to produce multistable systems, with broad potential implications for reconfigurable systems and microrobotics. Finally, we combine the effects of the NLC elastic energy field and of an external field (gravity) to demonstrate fine-tuning of the particles' sensitivity to the size of their docking sites.

## Results

**Molding the energy landscape.** To mold the elastic energy landscape near a curved boundary with geometrical parameters

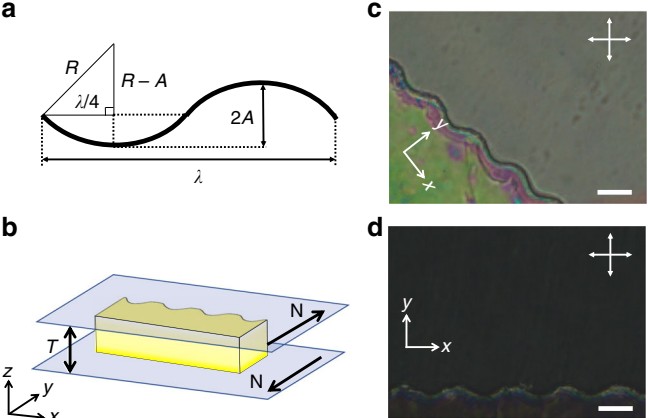

**Fig. 1** Schematic of experiment. **a** Schematic of the wall shape with relevant parameters: radius of curvature $R$, amplitude $A$, and wavelength $\lambda$. **b** Schematic of the experimental setup (**N** denotes rubbing direction, $T$ denotes thickness of the cell). **c, d** Cross polarized images of liquid crystal near the wavy wall with the long axis either (**c**) at a 45° angle to the polarizer or (**d**) perpendicular to the polarizer. The scale bars are 20 μm

defined in Fig. 1a, we fabricate long, epoxy resin strips using standard lithographic techniques to form wavy structures (Fig. 1b). These structures are placed between two parallel glass slides, separated by distance $T$, with planar anchoring oriented perpendicular to the strip (see Methods for details and parameters) to form a cell within which the NLC is contained. This cell is filled by capillarity with a suspension of colloids in the NLC 4-cyano-4'-pentylbiphenyl (5CB) in the isotropic phase, and subsequently quenched into the nematic phase ($T_{NI} = 34.9$ °C). The alignment of a colloid-free cell is examined under crossed polarizers (Fig. 1c, d), which shows that the bulk liquid crystal is defect-free. The much brighter texture at 45°–135° (Fig. 1c) compared to the 0°–90° (Fig. 1d) also shows good planar alignment along the $y$ direction. The defects visible in Fig. 1c, d are only in the thin NLC film squeezed between the top of the wavy wall and the confining glass, a region which is not accessible to the colloids.

Colloid migration in the cells is observed with an optical microscope from a bird's-eye view. For the larger colloids, as expected, strong confinement between the glass slides stabilizes the Saturn ring configuration[26], with a disclination line encircling the colloid. Smaller colloids, which experience weaker confinement, adopt the dipolar structure where a colloid is accompanied by a topological point-like defect often called a hedgehog. Particles are equally repelled by elastic interactions with the top and bottom glass slides, whose strength dominates over the particles' weight, so gravity plays a negligible role in our system[27] when the $z$ axis of our experimental cell is vertical. When observed through the microscope, this configuration forms a quasi-2D system in the $(x,y)$ plane, where $y$ is the distance from the base of a well in the direction perpendicular to the wall. Unless otherwise specified, when reporting colloid position, $y$ denotes the location of the colloid's center of mass (COM).

The wavy wall forms a series of hills and wells, with amplitude $2A$ measured from the base of the well to the highest point on a hill. Because of strong homeotropic anchoring at the wavy wall, these features impose zones of splay and bend in this domain. In particular, the valleys are sites of converging splay, the hills are sites of diverging splay, and the inflection points are sites of maximum bend. The wavelength of the structure $\lambda$ can be expressed in terms of the radius of curvature $R$ and the amplitude $A$: $\lambda = 4R\sqrt{\frac{A}{R}\left(2 - \frac{A}{R}\right)}$ (Fig. 1a). Therefore, $\lambda$ and $R$ are not

independent for fixed $A$. Different aspects of the colloid-wall interaction are best described in terms of one or the other. For example, the range of the distortion is discussed in terms of $\lambda$, and the splay field near the well is described in terms of $R$. Throughout this study, unless specified otherwise, $2A = 10\ \mu m$. The gentle undulations of this wall deform the surrounding director field, but do not seed defect structures into the NLC. We demonstrate control over colloidal motion within the energy landscape near this wall. In addition, we use Landau-de Gennes (LdG) simulation of the liquid crystal orientation to guide our thinking. Details of the simulation approach can be found in the Methods section.

**Attraction to the wall**. To determine the range of interaction of a colloid with undulated walls of differing $\lambda$, a magnetic field is used to move a ferromagnetic colloid (radius $a = 4.5\ \mu m$) to a position $y$ far from the wall and $x$ corresponding to the center of the well. The magnet is rapidly withdrawn, and the colloid is observed for a period of 2 min. If the colloid fails to approach the wall by distances comparable to the particle radius within this time, the colloid is moved closer to the wall in increments of roughly a particle radius until it begins to approach the wall. We define the range of interaction $H^{\star}$ as the maximum distance from the base of the well at which the colloid starts moving under the influence of the wall (Fig. 2). In these experiments, the Saturn ring defect was sometimes pinned to the rough surface of the ferromagnetic particles. To eliminate this effect, these experiments were repeated with homeotropic magnetic droplets with a smooth interface whose fabrication is described in the Methods section. The results did not change. A typical trajectory is shown in Fig. 2a in equal time step images ($\Delta t = 125\ s$). For small $\lambda$ (i.e., $\lambda \lesssim 40\ \mu m$), $H^{\star}$ increases roughly linearly with $\lambda$. However, at larger $\lambda$, the range of interaction increases only weakly. A simple calculation for the director field near a wavy wall in an unbounded medium in the one elastic constant approximation and assuming small slopes

predicts that the distortions from the wall decay over distances comparable to $\lambda$[24]. However, for $\lambda$ much greater than the thickness of the cell $T$, confinement by the top and bottom slides truncates this range (see Supplementary Note 1 and Supplementary Figure 1), giving rise to the two regimes reported in Fig. 2b: one that complies with the linear trend and one that deviates from it. A similar shielding effect of confinement in a thin cell was reported in the measurements of interparticle potential for colloids in a sandwich cell[28].

The colloid moves toward the wall along a deterministic trajectory. Furthermore, it moves faster as it nears the wall (Fig. 2c), indicating steep local changes in the elastic energy landscape. This motion occurs in creeping flow (Reynolds number $Re = \rho v a / \eta \approx 1.15 \times 10^{-8}$, where $\rho$ and $\eta$ are the density and viscosity of 5CB, respectively, and $v$ is the magnitude of the velocity of the colloid). The energy $U$ dissipated to viscous drag along a trajectory can be used to infer the total elastic energy change; we perform this integration and find $U \sim 5000\ k_{B}T$. In this calculation, we correct the drag coefficient for proximity to the wavy wall according to Ref. [29] and for confinement between parallel plates according to Ref. [30] (see Ref. [24] for more details). The dissipation calculation shows that gradients are weak far from the wall and steeper in the vicinity of the wall. The elastic energy profile found from LdG simulation as a function of particle distance from the base of the well is consistent with these observations (Supplementary Figure 2). The particle finds an equilibrium position in the well. At larger distances from the wall, the energy increases first steeply, and then levels off (Supplementary Figure 2). For wide wells ($\lambda > 15a$), the energy gradient in $x$ near the wall is weak, and the drag is large. In this setting, the colloid can find various trapped positions, and introduce error to the energy calculation. Therefore, the trajectory is truncated at around $y = 15\ \mu m$ from contact with the wall.

**Equilibrium position**. The wall shape also determines the colloid's equilibrium position $y_{eq}$, i.e., the distance between the colloid's COM and the bottom of the well. In fact, we show that the particles do not always dock very close to the wall. Rather, they find stable equilibrium positions at well-defined distances from contact with the hills and wells. We probe this phenomenon by varying colloid radius $a$ and well radius of curvature $R$ (Fig. 3a). At equilibrium, $y_{eq}$ is equal to $R$. That is, the colloid is located at the center of curvature of the well (Fig. 3b, c). In this location, the splay of the NLC director field from the colloid matches smoothly to the splay sourced by the circular arc that defines the well. As $R$ increases, this splay matching requirement moves the equilibrium position of the colloid progressively away from the wall.

However, for wide wells with $R \gg 2a$, the elastic energy from the wall distorts the Saturn ring, displacing it away from the wall (Fig. 3d, e). When this occurs, the equilibrium position of the colloid is closer to the wall. For all such colloids, the height of Saturn rings (Fig. 3a crosses: $y = y_{def}$) and that of the COM of the particles (Fig. 3a open circles: $y = y_{eq}$) do not coincide. Specifically, the particle moves closer to the wall, and the disclination line becomes distorted, i.e., the Saturn ring moves upward from the equator of the particle so that the particle-defect pairs become more dipole-like (Fig. 3g, h). For comparison, we plot the COM of particles with point defects sitting near the wall (Fig. 3f). We observe that, when the colloid radius is similar to the radius of the wall ($R/a \approx 2$), there is a similar "splay-matching" zone for the dipoles; however, as we increase $R/a$, the behavior changes. In this regime, the dipole remains suspended with its hedgehog defect at a distance of roughly $y_{def}/a = 3$ from the base of the well for wells of all sizes. The equilibrium distance of

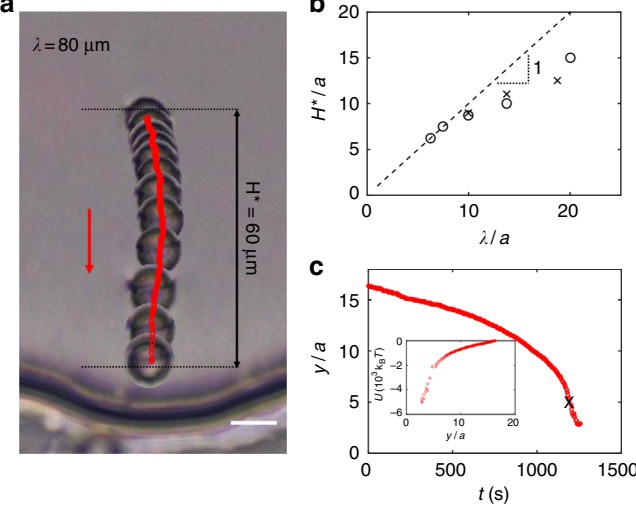

**Fig. 2** Colloid-wall interaction range vs. wavelength $\lambda$. A ferromagnetic homeotropic colloid with a Saturn ring defect is used to establish the range of interaction $H^{\star}$ of the colloid with the wall. **a** An equal time step ($\Delta t = 125$ s) image is shown for the case $\lambda = 80\ \mu m$, $H^{\star} = 60\ \mu m$. **b** Range of interaction $H^{\star}$ vs. the wavelength of the feature $\lambda$ for homeotropic droplets (open circles) and homeotropic colloids (crosses). **c** The position of the particle $y$ with respect to time $t$. Inset: Energy dissipated to viscosity along a particle trajectory $U$ with respect to the particle position $y$. The cross shows where we truncate the trajectory for integration along the path to infer the dissipation. The scale bar is 10 $\mu m$

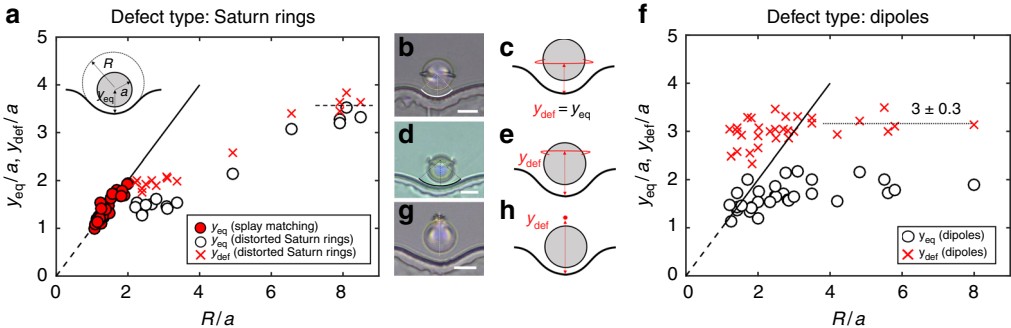

**Fig. 3** Particle-wavy wall interactions mechanisms. $y_{eq}$ and $y_{def}$ measure the equilibrium distance relative to the bottom of the wells of the wavy wall for the COM of the colloid and the defect. **a** Filled red circles denote splay matching cases, where the Saturn ring sits at the equatorial position ($y_{def}/a = y_{eq}/a$). Crosses denote location of distorted Saturn rings, $y_{def}/a > y_{eq}/a$. Open circles indicate the height of the center of mass (COM) of the colloid. The dotted line denotes flat wall limit. Inset: Schematic of system geometry. **b, c** Experimental bright field microscopy image and schematic of splay matching. **d, e** Experimental bright field microscopy image and schematic of distorted Saturn ring. **f** Heights of the center of mass (COM, open circles) and hedgehog defects (crosses) of the colloid with dipole defects. **g, h** Experimental bright field microscopy image and schematic of dipoles and their defects. The scale bars are 10 μm

particles with distorted Saturn rings (Fig. 3a open circles) is intermediate between equilibria for particles with undistorted Saturn rings and colloids in dipolar configurations with point defects. LdG simulation corroborates the finding that dipoles and quadrupoles equilibrate at different distances from the wall, and that the particles with point defects sit deeper in the well than those with Saturn ring (Supplementary Figure 3).

A colloid positioned directly above a well moves down the steepest energy gradient, which corresponds to a straight path toward the wall. The energy minimum is found when the particle is at a height determined by $R/a$, consistent with our experiments (Fig. 3b). We also note that at $R/a = 7$, we find $y_{COM}/a = 3.5$, which corresponds to the equilibrium distance of colloids repelled from a flat wall. However, even at these wide radii, the elastic energy landscape above the undulated wall differs significantly from the repulsive potential above a planar boundary, which decays monotonically with distance from the wall[31]. For colloids above the wide wells, energy gradients in the $y$ direction are small, but gradients in the $x$ direction are not. As a result, particles migrate laterally and position themselves above the center of the wells. We have postulated and confirmed the splay matching mechanism to be the driving force of the colloid docking. We expect that by using a liquid crystal that has different elastic constants, we can enhance or suppress this effect. For example, for a LC with $K_{11} > K_{33}$, the colloids will preferentially sit closer to the wall to favor bend distortion over splay.

**Quadrupole to dipole transition.** For micron-sized colloids in an unbounded medium, the dipole is typically the lowest energy state[32]; electrical fields[33], magnetic fields[34] or spatial confinement[26] can stabilize the Saturn ring configuration. In prior research, we showed that a colloid with a Saturn ring defect, stabilized by confinement far from the wavy wall, became unstable and transformed into a dipolar structure near the wavy wall[24]. However, in those experiments, the transformation occurred very near the wall, where the dynamics of the colloid and surrounding liquid crystal were strongly influenced by the details of wall-particle hydrodynamic interactions and near-wall artifacts in the director field. Here, to avoid these artifacts, we use wells with a smooth boundary where $R > a$ and amplitude $A > a$ (specifically, $A = R = 15$ μm and $\lambda = 60$ μm, or $A = R = 25$ μm and $\lambda = 100$ μm). These wells are deeper and are best described as semicircular arcs with rounded corners.

We exploit these wider wells to position a colloid with a companion Saturn ring several radii above the wall. The elastic

energy field distorts the Saturn ring, and drives a gentle transition to a dipolar defect configuration, as shown in Fig. 4a in time lapsed images. The location of the colloid $y$ and the evolution of the polar angle of maximum deflection $\theta$ are tracked and reported in Fig. 4b. This transition is not driven by hydrodynamics; the Ericksen number in this system is $Er = 8 \times 10^{-4}$, a value two orders of magnitude lower than the critical value $Er = 0.25$ at which a flow-driven transition from quadrupole to dipole occurs[35].

The confinement from the top and bottom glass stabilizes the Saturn ring. The wavy wall, however, exerts an asymmetrical elastic energy gradient on the Saturn ring, displaces it away from that wall, and ultimately destabilizes this configuration. Once the transition to dipole has taken place, re-positioning the particle away from the wall with a magnetic field does not restore the Saturn ring (Supplementary Movie 2).

Previously, Loudet and collaborators[36] studied the transition of a colloid with a Saturn ring defect to a dipolar configuration in an unbounded medium, prompted by the fast removal of the stabilizing electric field. Although these two sets of experiments take place in very different physical systems (confined vs. unconfined, withdrawal of an electric field vs. an applied stress field via boundary curvature), the slow initial dynamics and the total time of transition are common features shared by both (Fig. 4c, d).

The dynamics of the transition are reproducible across particles of different sizes (Fig. 4e), and across additional runs with different sized walls (Supplementary Figure 4), and even in the case where debris is collected by the topological defects on the way. However, Loudet et al. observed a propulsive motion attributed to back flow from reorientation of director field in the direction opposite to the defect motion. In our system, the motion is smooth and continuous as the colloid passes through the spatially varying director field. Furthermore, the velocity of the droplet decreases right after transition; we attribute this, in part, to the change in the drag environment (Fig. 4b and Supplementary Figure 4b)

There are cases in which the transition does not occur; rather, the Saturn ring remains distorted. In such cases the polar angle then ranges from $\theta = 103°$ to $130°$. For polar angles larger than $130°$, the transition always occurs, indicating that this is the critical angle for the transition. This value, however, differs from that measured in Ref. [37]. This difference may be attributed to the differing confinement of the cell. Differences in anchoring and elastic constants may also play a role.

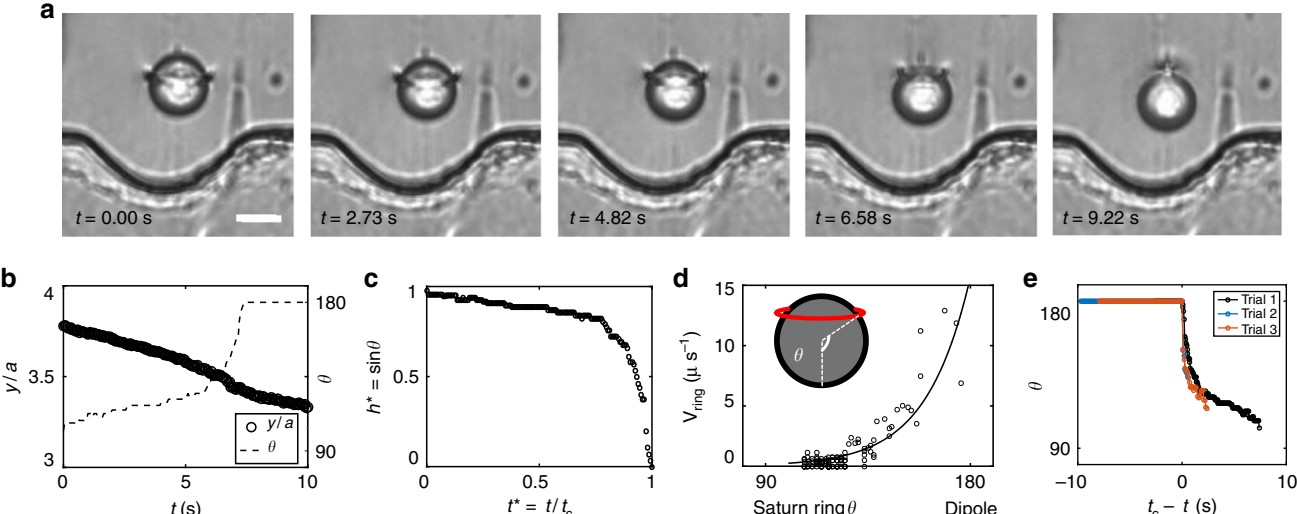

**Fig. 4** Dynamics of the quadrupole to dipole transition. **a** Time-lapse image of a Saturn ring transforming to a dipole at a metastable position remotely from the wall defined by the elastic energy field (see Supplementary Movie 1). The scale bar is 10 μm. **b** The $y$ location of the colloid's center of mass (COM) and evolution of the polar angle $\theta$ during the transition. Initially, the colloids assume the $\theta = 90°$ (Saturn ring) configuration, which gradually evolves to $\theta = 180°$ as the COM continuously moves towards the wall. After the transition to a dipolar configuration, the particle approaches the wall. **c, d** Reduced ring size and velocity from our system reveal similar dynamics of transition as shown in Fig. 2 in Ref. [36]. The solid line serves as guide to the eye. **e** $\theta$ vs. $t_c - t$ plot shows three experimental runs of transition in similar geometry. In **b-d**, $t_c$ is the time at which $\theta = 90°$

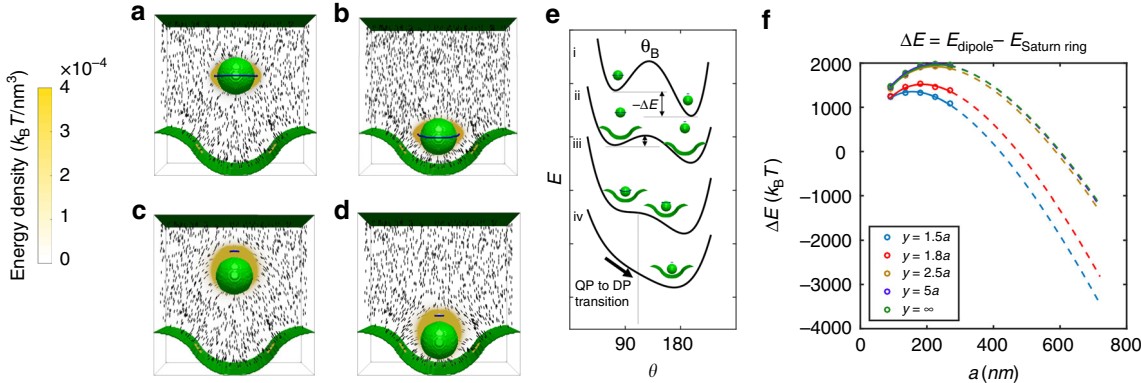

**Fig. 5** Simulated energy density for dipole and quadrupole near a wavy boundary. By exploring the energy for colloids in dipole (DP) and Saturn ring (QP) configurations at various positions above the well for fixed colloid size and wavy wall geometry, the equilibrium heights for the Saturn ring are found. **a** A Saturn ring located at the reference state far from the wall (state 1, $y = 5a$). **b** A Saturn ring located at its equilibrium location (state 2, $y = 1.8a$), a decrease of 203.5 $k_B T$ from state 1. **c** A dipole located at the reference state far from the wall (state 1, $y = 5a$). **d** A dipole located at its equilibrium location (state 2, $y = 1.5a$), with an energy decrease of 585.01 $k_B T$ from state 1. **e** Schematic representation of the total energy of the system $E$ vs. the reaction coordinate $\theta$ for several distances $y$ from the well, changing from far from the well to close to the well (i through iv) as $E$ decreases. The presence of the well shifts the angle of the energy barrier's maximum to the right (increasing $\theta$) and decreases the energy barrier until it is eliminated as the particle moves closer to the wall. **f** The energy of the dipole and quadrupole are calculated for systems of different size (colloid radius $a = 90, 135, 180, 225, 270$ nm; the simulation box and the walls are scaled accordingly). The energy difference between quadrupole and dipole ($\Delta E = E_{dipole} - E_{Saturn\ ring}$) is plotted against $a$. Circles denote simulation results, solid lines are fitted to forms suggested by scaling arguments, dotted line are extrapolations based on these fits

**Quadrupoles and dipoles in simulation**. In deeper wells ($A > a$), the polar angle increases as the colloid migrates into the well. LdG simulation reveals that, in the dipolar configuration, there is less distortion in the director field near the colloid owing to bend and splay matching, and that it is indeed more favorable for a colloid with dipolar defect to be located deep within the well (Fig. 5a–d). In simulation, we compute the energy of a colloid both far (state 1: $y = 5a$, reference state) and near the wavy wall (Fig. 5a–d) to locate the equilibrium site for both the Saturn ring and dipolar configurations (state 2: $y = 1.8a$ and $y = 1.5a$, for Saturn ring and dipolar configuration, respectively). Details of this calculation are given in Methods. Using the same geometrical parameters and anchoring strength for the LdG numerics, we stabilize a dipolar

configuration by initializing the director field by the dipolar far-field ansatz[38]. While colloids in both configurations decrease their energy by moving from state 1 to state 2, the decrease in energy is 2.9 times greater for the dipolar case (Fig. 5c, d). This change is determined by differences in the gradient free energy, corresponding to reduced distortion in the nematic director field.

Stark[39] argues that the stabilization of a Saturn ring under confinement occurs when the region of distortion becomes comparable to or smaller than that of a dipole, assuming the same defect energy and energy density. Yet this argument does not apply here because the presence of the wavy wall strongly alters the energy density at various regions in the domain (Fig. 5a–d).

Since this reorganization occurs in creeping flow and at negligible Erickson number, it occurs in quasi-equilibrium along the reaction coordinate. In principle, this suggests that insight can be gained into the transition energy between the two states by simulating the equilibrium value for $\theta$ and the corresponding system energy $E$ for a colloid Saturn ring configuration at various fixed heights above the wall. We can consider the polar angle $\theta$ and the director field as our "reaction coordinate" to characterize the transition between the Saturn ring state ($\theta = 90°$) and the dipolar state ($\theta = 180°$). As shown schematically in Fig. 5e, an energy barrier exists between these two states far from the wall. The experiment indicates that this barrier is eliminated by the elastic energy field of the wall as the colloid approaches the well for certain geometries. Unfortunately, we are limited in how thoroughly we can explore this concept in simulation. The particle radii in our experiments are too large to be accurately reproduced, and must be re-scaled with caution, owing to the correlation length, which does not scale with system size.

In particular, our simulations are limited to particle radii for which the dipole is more costly than the Saturn ring everywhere in the domain, i.e., far from the wall and in its vicinity. Our experiments, recall, are performed with particle radii for which the dipole is the stable state, and the Saturn ring is metastable. Thus, direct calculations cannot yet capture the manner in which the energy landscape near the wall eliminates the energy barrier between and Saturn and dipole configuration, driving the transformation. Rather, direct calculations of system energy $E$ vs. $\theta$ for small colloids with stable Saturn rings simply show an energy minimum and an equilibrium ring displacement at their equilibrium height above the well (Supplementary Figure 5).

We can compare the system energy for quadrupolar and dipolar configurations by computing $\Delta E = E_{\text{dipole}} - E_{\text{Saturn ring}}$ (Fig. 5f, Supplementary Figure 6). This quantity is always positive for colloidal radii accessible in simulation. By moving closer the the wall, however, $\Delta E$ decreases (Fig. 5a–d, f). To explore how $\Delta E$ scales with colloid radius, we calculate $\Delta E$ in systems of similar geometries in which all length scales are increased proportionally with $a$ for a range of values (colloid radius $a = 90, 135, 180, 225, 270$ nm) (Fig. 5f, Supplementary Figure 6). The total energy consists of two parts, the phase free energy which captures the defect energy, and the gradient free energy which captures the distortion of the field. The hedgehog defect does not grow with the system size, while the Saturn ring grows with the linear dimension of the system. Thus, the difference in the phase free energy $\Delta E_{\text{phase}}$ between dipole and quadrupole is always linear in $a$ (Supplementary Figure 6a). However, the gradient free energy $\Delta E_{\text{gradient}}$ has more complex scaling, with a part that scales linearly in $a$ and a part that scales as a $\log(a)$[38]. Simulated values for $\Delta E_{\text{gradient}}$ are fitted to such a form $k_1 a + k_2 a \log(a) + k_3$, Supplementary Figure 6b).

The sum of these two ($\Delta E = \Delta E_{\text{phase}} + \Delta E_{\text{gradient}}$) for different $y$ values is presented in Fig. 5f (circles: simulated results; solid line: fit; dotted lines: extrapolations to micron-sized particles). Note that for large $a$ values, comparable to those in experiment, the linear-logarithmic form fitted to $\Delta E_{\text{gradient}}$ is linear in $a$. Extrapolation of $\Delta E$ according to the scaling arguments presented above suggests that $\Delta E$ becomes negative for large enough $a$. In this limit, the dipole becomes the stable configuration everywhere in the domain, in agreement with experiment. Furthermore, this suggests that, as a particle moves closer to the wall, the dipolar configuration is more favored.

These results show that the distortion field exerted by the wavy boundary can be considered as an external field, in some ways analogous to external electrical, magnetic or flow fields. However, the spatial variations in the elastic energy landscape and its dependence on boundary geometry allow gentle manipulations of

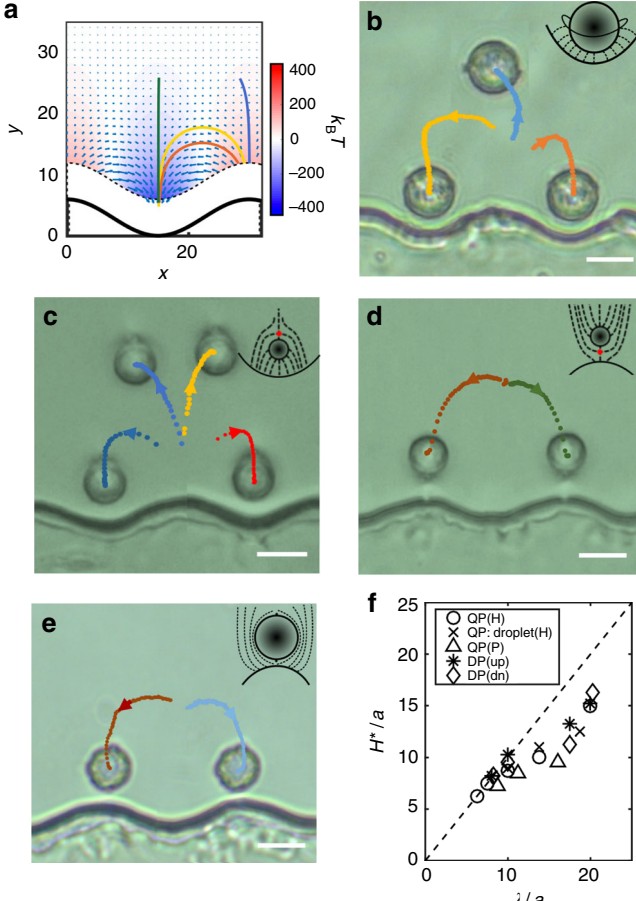

**Fig. 6** Multiple states and reconfigurable docking. **a** Elastic energy field and the resulting forces for colloids near the wall in a Saturn ring configuration. **b–e** Particle paths are illustrated by points that indicate particle COM position over time; time step $\Delta t = 5$ s between neighboring points. The colored dots denote: **b** three representative trajectories (out of 28) of a colloid with Saturn ring defect. **c** Four representatives trajectories (out of 12) of an upward-oriented dipole. **d** Two representative trajectories (out of 11) of a downward-oriented dipole. **e** Two representative trajectories (out of 14) of a planar-anchoring colloid with two boojums released between two neighboring wells. Insets: schematics of colloids with respective defect types. The scale bars are 10 μm. **f** The range of interaction $H^*$ as a function of $\lambda$ is similar for homeotropic (H) and planar (P) anchoring, for hedgehog (DP) and Saturn ring (QP) defects, and for solid colloids and droplets

colloids and their defects that are not typically afforded by those other fields.

## Multiple paths diverging from unstable points

The elastic energy field in the vicinity of the wall was simulated by placing the COM of a colloid in a Saturn ring configuration at different locations $(x, y)$. The reference energy is evaluated at $(\lambda/2, \lambda)$, where, recall, $\lambda$ is the wavelength of the periodic structure of the wall (Fig. 6a). The energy in the color bar is given in $k_B T$ for a colloid 54 nm in radius. The vectors in this figure show local elastic forces on the particle, obtained by taking the negative gradient of the elastic energy field. The solid curves indicate a few predicted trajectories for colloids placed at different initial positions in the energy landscape. (Further details of how this energy landscape is generated can be found in Supplementary Note 3 and Supplementary Figure 7). In the preceding discussions, we have focused on attractive particle-wall interactions and associated stable or metastable equilibria, which correspond to the energy

minima (blue) above the well. However, the location directly above a hill is an unstable point. When colloids are placed near this location using an external magnetic field, they can follow multiple diverging paths upon removal of the magnetic field. The particular paths followed by the colloid depend on small perturbations from the unstable point. Trajectories are computed by taking a fixed step size in the direction of the local force as defined by the local energy gradient (Fig. 6a).

In our experiments, amongst 28 trials using an isolated homeotropic colloid with a Saturn ring, the colloid moved along a curvilinear path to the well on its left 11 times, to the well on its right ten times and was repelled away from the peak until it was approximately one wavelength away from the wall seven times. Three sample trajectories are shown in Supplementary Movies 3–5. These trajectories are also consistent with the heat map in Fig. 6a. The numerically calculated trajectories (Fig. 6a) and their extreme sensitivity to initial position are in qualitative agreement with our experimental results (Fig. 6b). Thus, small perturbations in colloid location can be used to select among the multiple paths.

So far we have primarily discussed colloids with Saturn ring defects, but we can also tailor unstable points and attractors for dipolar colloids, and find important differences between the behavior of colloids attracted to wells and those attracted to hills. For example, a dipole pointing away from the wall (Fig. 6c) behaves like a colloid with companion Saturn rings in several ways. Both are attracted over a long range to equilibrate in wells, and both have unstable points above hills. Also, when released from this unstable point, both defect structures can travel in three distinct directions (left, right, and away from the wall, Fig. 6c). On the other hand, dipoles pointing toward the wall (Fig. 6d) behave differently. They are attracted to stable equilibria near hills, and are unstable near wells. Interestingly, when released from a point near a well, these colloids can travel only toward one of the adjacent hills. That is, there is no trajectory above the well that drives them in straight paths away from the wall.

Finally, we observed the behavior of colloids with planar molecular anchoring, which form two topologically required "boojums", surface defects at opposing poles[40]. They behave similarly to downward-orienting dipoles (Fig. 6e); they equilibrate near the hills, in accordance with the simulations of Ref. [41], and they follow only two sets of possible paths when released from unstable points above a well. The ability to drive particle motion with a gently undulating wall is thus not limited to colloids with companion Saturn rings; the wall also directs the paths of dipolar colloids with homeotropic anchoring and colloids with planar anchoring, decorated with boojums.

These results indicate that the range of repulsion differs for hills and wells. This is likely related to the differences in the nematic director field near these boundaries. While converging splay field lines are sourced from the well, divergent splay field lines emanate from the hill. Both fields must merge with the oriented planar anchoring far from the wall. As a result, hills screen wells better than wells screen hills. The ranges of interaction for various colloid-defect configurations are summarized in Fig. 6f; while colloids with each defect structure have distinct equilibrium distances from a flat wall (Supplementary Figure 8), the range of interaction between colloids and wavy walls follows a similar trend independent of the topological defects on the colloid (Fig. 6f).

**Extending the range of interaction by placing wavy walls across from each other**. Thus far, we have discussed instances of colloids of different defect structures diverging along multiple paths from unstable points near wavy walls. These features can be used to launch the colloid from one location to another, propelled by the elastic energy field. To demonstrate this concept, we arranged two wavy walls parallel to each other with the periodic structures in phase, i.e., the hills on one wall faced valleys on the other (Fig. 7a). For wall-to-wall separations more than $2\lambda$, colloids with Saturn rings docked, as expected (Fig. 7b). For wall-to-wall separations less than $2\lambda$, a colloid, placed with a magnetic field above the peak on one wall, was guided by the NLC elastic energy to dock in the valley on the opposite wall (Fig. 7c), thus effectively extending its range of interaction with the second wall (Supplementary Movie 6). In the context of micro-robotics, such embedded force fields could be exploited to plan paths for particles to move from one configuration to another, guided by a combination of external magnetic fields and NLC-director field gradients.

We can also exploit wall-dipole interactions to shuttle the colloid between parallel walls. For walls positioned with their wavy patterns out-of-phase (Fig. 7d, Supplementary Movie 7), dipoles with point defect oriented upwards are repelled from initial positions above hills on the lower wall and dock on the hill on the opposite wall. However, for walls with their patterns in phase, dipoles with defects oriented downwards released from an initial position above a well dock either at an adjacent hill on the same wall (Fig. 7e, Supplementary Movie 8), or in an attractive well on the opposite wall (Fig. 7f, Supplementary Movie 9).

**"Goldilocks" or well-selection for colloids in motion**. Particles in motion can select preferred places to rest along the wavy wall. Wells with different wavelengths create energy gradients that decay at different, well-defined distances from the wall. Placing wells of different sizes adjacent to each other offers additional opportunities for path planning. In one setting that we explore, a colloid can sample multiple wells of varying sizes under a background flow in the $x$ direction. We followed a colloid moving under the effect of gravity. The sample was mounted on a custom-made holder that can be tilted by an angle $\alpha$ (Fig. 8a, b) within a range between 10° and 20° so that the colloid experiences a body force in the $x$ direction. We have verified in independent experiments that, without the wall, the particle moves at a constant velocity due to balance of drag and gravity. In the presence of the wavy wall, the particle's trajectory is influenced by the energy landscape there. We first describe the particle paths over a series of periodic wells, and then describe motion for wells of decreasing wavelengths.

Docking or continued motion in the cell is determined by a balance between the body force that drives $x$-directed motion and viscous forces that resist it, the range and magnitude of attractive and repulsive elastic interactions with the wall, and viscous drag near the wall. If the particle moves past the well in the $x$ direction faster than it can move toward the wall, it will fail to dock. However, if interaction with the well is sufficiently pronounced to attract the particle before it flows past, the particle will dock.

For a tilted sample with a wavy wall of uniform wavelength ($\lambda = 70\,\mu m$), colloids initially close enough to the wall dock into the nearest well (Fig. 8c, $V_x = 0.01\,\mu m\,s^{-1}$, Supplementary Movie 10). Far from the wall, the colloids do not dock. However, the influence of the wall is evident by the fact that the colloids do not remain at a fixed distance from the wall. Rather, the distance from the wall varies periodically, and this periodic motion has the same wavelength as the wall itself (Fig. 8d, $V_x = 0.06\,\mu m\,s^{-1}$, Supplementary Movie 11).

To simulate the forces on the particle, a particle is placed at different locations near a wall, and the energy of the system is calculated (as detailed in Supplementary Note 3). Gradients in this energy capture the forces on the colloid owing to the distortions of the director field at each location. A uniform body

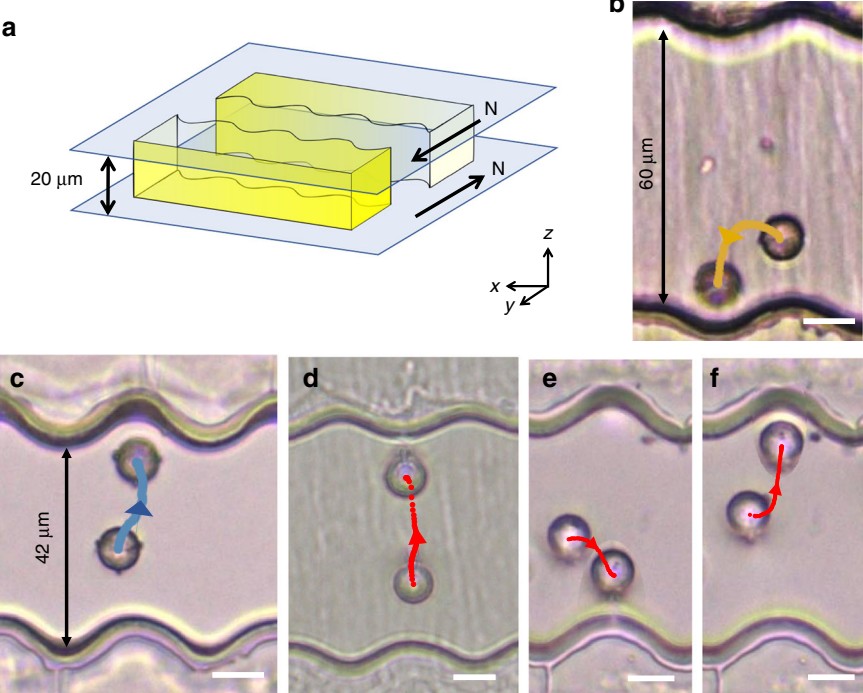

**Fig. 7** Repulsion and bistable docking of dipoles and Saturn rings. **a** Schematic of two parallel walls with a gap comparable to $2\lambda$ between them. **b, c** A magnetic particle with a Saturn ring defect, placed near a hill, with wall to wall separation (**b**) 60 μm and (**c**) 42 μm. In **b**, the particle is more attracted to the wall on the same side. In **c** the particle is repelled from the hill, and traverses the separation between walls to dock in the well on the opposite side ($2a = 9$ μm). **d–f** Behaviors of the dipoles. The waves of the wall are either out of phase with hill to hill configuration such as in **d** or in phase with hill to valley such as in **e, f**. The scale bars are 10 μm

force in the $x$ direction is then added on the colloid to find the trajectories. We simulated the trajectories for various initial loci. We find two outcomes: for strong $x$-directed force and/or far from the wall, the particle follows a wavy path (Fig. 8e, yellow trajectory); for weak $x$-directed force and near the wall, the particle docks (Fig. 8e, red and green trajectories). A particle slows down right before the hill and speeds up as it approaches the next well. This velocity modulation can be attributed to the interaction with the splay-bend region, similar to particles moving within an array of pillars[11]. Our experiments and simulations are in good agreement, showing both behaviors.

However, a different behavior is observed when we modulate the wavelength of the wavy wall, by placing wells adjacent to each other with different wavelength as defined in Fig. 1a. As a particle travels past successive wells of decreasing wavelengths ($\lambda = 70, 60, 50, 40$ μm, Supplementary Movie 12), the particle moves in the $y$ direction, closer to the wells, until it eventually is entrained by a steep enough attraction that it docks (Fig. 8f, $V_x = 0.09$ μm s$^{-1}$). This particle, like Goldilocks, protagonist of a beloved children story, finds the well that is "just right". Simulation of two wells with different wavelengths and a superimposed force confirms these results: we can achieve an additional state not possible with the uniform well, i.e. a wavy trajectory that descends and docks (Fig. 8g, yellow trajectory). The slight energy difference between wells of different wavelength underlies the "Goldilocks" phenomenon. Since the energy landscape defines zones of strong bend and splay, the ratio between the elastic constants $K_{11}$ and $K_{33}$ is important in determining the particle paths. Such interactions open interesting avenues for future studies, in which the rates of motion owing to elastic forces and those owing to applied flows are tuned, and the trapping energy of the docking sites are tailored, e.g., for colloidal capture and release.

## Discussion

The development of robust methods to drive microscopic objects along well-defined trajectories will pave new routes for materials assembly, path planning in microrobotics and other reconfigurable micro-systems. Strategies developed within NLCs are one means to address these needs. Since the strategies developed in liquid crystals depend on topology, confinement, and surface anchoring, which can be manipulated by changing surface chemistry or texture on colloids with very different material properties, they are broadly applicable across materials platforms. We have developed controllable elastic energy fields in NLCs near wavy walls as a tool to manipulate the ranges of attraction and to define stable equilibiria. We have also exploited elastic energy fields to drive transitions in topological defect configurations. The near-field interaction between the colloid and the wall rearranges the defect structure, driving a transition from the metastable Saturn ring configuration to the globally stable dipolar configuration for homeotropic colloids.

We account for this transformation by means of an analogy between confinement and an external applied field. However, the gentle elastic energy field allows us to access metastable states. As these defect sites are of interest for molecular and nanomaterials assembly, the ability to control their size and displacement will provide an important tool to improve understanding of their physico-chemical behavior, and potentially to harvest hierarchical structures formed within them.

Furthermore, we have developed the concept of repulsion from unstable points as a means to dictate paths for colloids immersed within the NLCs. We have identified unstable sites from which multiple trajectories can emerge, and have used these trajectories to propel particles, demonstrating the multistability made possible by the wavy wall geometry. Finally, we have demonstrated the

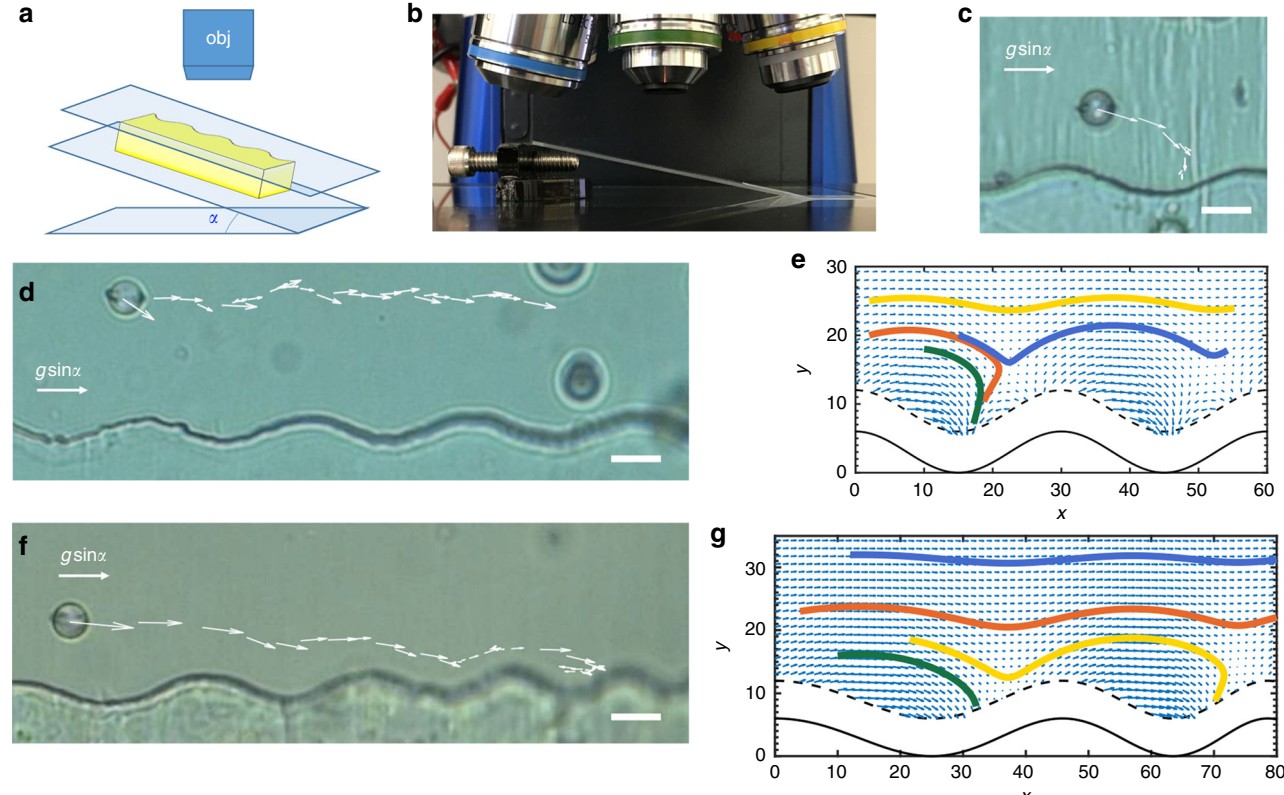

**Fig. 8** A "Goldilocks" colloid in motion docks in a preferred well. **a** Schematic of the experimental setup with tilt angle $\alpha$ to harness gravity to drive colloid motion in $x$ direction. **b** Image of the setup which allows fine control of the tilt, and thus the $x$-component of gravity $g\sin\alpha$. **c** A colloid traveling with $x$-directed velocity $V_x = 0.01\ \mu m\ s^{-1}$ at tilt angle $\alpha = 11.2°$. **d** A colloid traveling at $V_x = 0.06\ \mu m\ s^{-1}$ at tilt angle $\alpha = 12.3°$. **e** Particle trajectories at various initial loci predicted for the sum of the elastic energy field and a gravitational potential energy gradient in $-x$ direction across uniform set of wells ($\lambda = 70\ \mu m$). **f** A colloid traveling at $V_x = 0.09\ \mu m\ s^{-1}$ at tilt angle $\alpha = 12.8°$ across wells of decreasing wavelengths ($\lambda = 70, 60, 50, 40\ \mu m$). **g** Particle trajectories predicted for the sum of the elastic energy field and a gravitational potential energy gradient in $-x$ direction. All colloids have $2a = 13\ \mu m$. The scale bars are 20 $\mu m$

Goldilocks concept, i.e., that wells of different wavelengths can be used to guide docking of particles moving in a superimposed flow or via an external force. These concepts lend themselves to actuation and path planning in reconfigurable systems.

## Methods

**Assembly of the cell**. We have developed a wavy wall confined between two parallel plates as a tool to direct colloid assembly. The wavy wall is configured as a bounding edge to the planar cell. The NLC cell and the walls were fabricated following the procedure in Ref. [24]. The procedure is briefly outlined here. The wavy walls are made with standard lithographic methods of SU-8 epoxy resin (Micro-Chem Corp.). The wells have wavelengths $\lambda$ ranging from 27–80 $\mu m$ and consist of smoothly connected circular arcs of radius $R$ between 7–40 $\mu m$. These strips, of thickness $T = 20$–28 $\mu m$, are coated with silica using silica tetrachloride via chemical vapor deposition, then treated with DMOAP (dimethyloctadecyl[3-(tri-methoxysilyl)propyl]). The wavy wall is sandwiched between two antiparallel glass cover slips, treated with 1% PVA (poly(vinyl alcohol)), annealed at 80 °C for 1 h and rubbed to have uniform planar anchoring. Once assembled, the long axis of the wall is perpendicular to the oriented planar anchoring on the bounding surfaces. We observed that in some LC cells the actual thickness was larger than expected, which we attribute to a gap above the strip. In those cases we noticed that some small colloids could remain trapped between the wavy strip and the top glass surface, so the effective thickness could be as large as 35–40 $\mu m$.

**Particle treatment and solution preparation**. We use the NLC 5CB (4-cyano-4'-pentylbiphenyl, Kingston Chemicals) as purchased. We disperse three types of colloids in the 5CB. The size and polydispersity of the colloids are characterized by measuring a number of colloids using the program FIJI. (1) $a = 7.6 \pm 0.8\ \mu m$ silica particles (Corpuscular Inc.), treated with DMOAP to have homeotropic anchoring. (2) $a = 4.3 \pm 0.4\ \mu m$ ferromagnetic particles with polystyrene core and coated with chrome dioxide (Spherotech, Inc.), treated with DMOAP, an amphiphile that imposes homeotropic anchoring, or with PVA for planar anchoring. (3) $a = 4.3$–8 $\mu m$ custom-made emulsion droplets where water phase was loaded with magnetic nanoparticles and crosslinked. The oil phase consisted of 5CB mixed with 2 wt%

Span 80. The water consisted of a 50:50 mixture of water loaded with iron oxide nanoparticle and a pre-mixed crosslinking mixture. The magnetic nanopowder iron (II, III) oxide (50–100 nm) was first treated with citric to make it hydrophilic. The crosslinking mixture was pre-mixed with HEMA (2-hydroxyl ethyl metha-crylate): PEG-DA (poly(ethylene glycol) diacrylate): HMP (2-hydroxyl-2-methyl-propiophenone) in 5:4:1 ratio. Water and oil phases were emulsified with a Vortex mixer to reach desired colloid size range. The two were combined in a vial treated with OTS (trichloro(octadecyl)silane) to minimize wetting of the wall by the water phase during the crosslinking process. All chemicals were purchased from Sigma Aldrich unless otherwise specified. The emulsion was crosslinked by a handheld UV lamp (UVP, LLC) at 270 nm at roughly power $P = 1\ mW\ cm^{-2}$ for 3 h. The emulsion was stored in a refrigerator for stability. Span 80 ensured that the liquid crystal-water interface would have homeotropic anchoring. The magnetic droplets are very poly-dispersed due to the emulsification process. However, when we compare their behavior with the silica and feromagnetic colloids, we only compare colloids and droplets of similar sizes.

**Imaging**. The cells form a quasi-2D system that is viewed from above. In this view, the wavy wall is in the plane of observation. The homeotropic colloids dispersed in the NLC are located between the top and bottom coverslips. These colloids are levitated away from both top and bottom surfaces by elastic repulsion[27]. The cell was imaged with an upright microscope (Zeiss AxioImager M1m) under magnification ranging from 20× to 50×. The dynamics of the colloid near the wavy wall are recorded in real time using optical microscopy. Additional information regarding the director field configuration is also gleaned using polarized optical microscopy.

**Application of a magnetic field**. The magnetic field was applied by using a series of 8 NdFeB magnets (K&J Magnetics, Inc.) attached to the end of a stick. The magnets was placed roughly 0.5 cm from the sample; the field applied is estimated to be roughly 40–60 mT, far below the strength required to reorient the NLC molecules, but sufficiently strong to overcome the drag and move magnetic dro-plets and particle in arbitrary directions.

**Numerical modeling by Landau-de Gennes (LdG) simulation**. Numerical modeling provides insight into the NLC-director field in our confining geometries. We use the standard numerical Landau-de Gennes (Q-tensor) approach[42] with a finite difference scheme on a regular cubic mesh. This approach is widely used to compute regions of order and disorder in bounded geometries through a global free energy minimization. The Q-tensor is a second-rank, traceless, symmetric tensor whose largest eigenvalue is the order parameter $S$ in the NLC. Using the Landau-de Gennes approach, at equilibrium, the 3-D director field and the locations of defect structures for a given geometry are predicted. The nematic director field, a headless vector field (i.e., $-n = n$), represents the average direction of an ensemble of molecules of size comparable to the correlation length at any point in the system. Defects are defined as the regions where the order parameter $S$ is significantly less than than the bulk value. The mesh size in our simulation is related to the correlation length in the NLC, and corresponds to 4.5 nm. Due to the difference in scale, the exact final configurations of numerics and experiment must be compared with caution; nevertheless, it is an invaluable tool to corroborate and elucidate experimental findings.

**Simulation geometry and parameters**. The geometry of the system, the boundary conditions, and elastic constants for the NLC are inputs to the numerical relaxation procedure. The one-constant approximation is used. Since we have a quasi-2D system, with the director field expected to lie in the plane of the wavy wall, the effect of changing the twist constant is expected to be weak in comparison to changing the splay and bend elastic constants. Specifically, the particle is bounded by walls with oriented planar anchoring separated by thickness $T = 4a$, unless otherwise specified. The effect of confinement with different $T$ values has been explored in detail in Supplementary Note 1 and Supplementary Figure 1. The anchoring at the boundary opposite of the wavy wall is set to zero, and that of the flat plates sandwiching the colloid and the wavy wall is set to oriented planar. The Nobili-Durand anchoring potential is used[43]. Because the size of simulation is much smaller, much stronger anchoring is applied. For most of our results, infinite anchoring strength is applied unless otherwise specified. To verify this assumption is valid, we simulate the particle placed at various distances from the wavy wall, centered above the well, and the anchoring strength is systematically varied. Under realistic anchoring strength ($10^{-3}$–$10^{-2}$ Jm$^{-2}$), the behavior of the energy of moving a colloid from near to far does not deviate much from that in the case of infinite anchoring (Supplementary Figure 9). As we decrease the anchoring further, the particle interacts with the well from a decreased range, and more weakly.

**Simulation of the dipoles**. To simulate dipoles, we vary the material constants $B$ and $C$ so that the core energy of the defect is 2.6x higher to compensate for the small system (details can be found in Supplementary Materials). In addition, we also use an initial condition with a dipolar configuration about the colloid: $\mathbf{n}(\mathbf{r}) = \hat{i} + PR_c^2 \frac{\mathbf{r} - \mathbf{r}_c}{|\mathbf{r} - \mathbf{r}_c|^3}$, where $R_c$ is the colloid radius, $\mathbf{r}_c$ is the location of the colloid center, $P = 3.08$ is the dipole moment, and $\hat{i}$ is the far-field director[38]. This expression is applied only in a sphere of radius $2R_c$ around $\mathbf{r}_c$.

**Numerical modeling by COMSOL**. To describe some aspects of the director field in the domain, we employ the common simplification in NLC modeling known as the one-constant approximation: $K_1 = K_2 = K_3 \equiv K$. If there is no bulk topological defect, then the director field is a solution to Laplace's equation $\nabla^2 \mathbf{n} = 0$, which can be solved by COMSOL separately for the two components $n_x$ and $n_z$, from which $n_y$ is obtained by the unit length restriction on $\mathbf{n}$. In COMSOL, this is easiest implemented by the Electrostatics Module. The model solves the equivalent electrostatic problem of $\nabla^2 V = 0$, which gives us $n_x$ and $n_z$. Customized geometry, such as the wavy wall, can be built with the geometry builder. We mesh the space with a triangular mesh and calculate the director field components; the results are then exported in grid form and post-processed in MATLAB.

**Code availability**. Code used for Landau-de Gennes and COMSOL numerical modeling approaches is available from the corresponding authors upon request.

## Data availability

The video and image data that support the findings of this study are available in figshare with the identifier https://doi.org/10.6084/m9.figshare.6840530

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

## Acknowledgements

This material is based upon work supported by, or in part by, the U. S. Army Research Laboratory and the U. S. Army Research Office under Grant W911NF1610288. We thank Dr. Sarah Hann for treatment of iron oxide NPs, Dr. Laura Bradley for useful discussion on synthesizing magnetic droplets, Prof. Ani Hsieh, Dr. Shibabrat Naik, Dr. Denise Wong, and Dr. Edward Steager for useful discussion on magnetic control and path planning.

## Author contributions

K.J.S, F.S., Y.L., and G.B. designed the project. Y.L. performed research. Y.L. and D.A.B. performed numerical modeling and theoretical analysis. K.J.S, F.S., Y.L., and D.A.B. wrote the manuscript.

## Additional information

**Competing interests:** The authors declare no competing interests.

