## [Peer Review File · Nature Communications]

Reviewers' comments:

Reviewer #1 (Remarks to the Author):

This manuscript presents a number interesting observations in terms of interactions between colloidal particles and wavy walls, in the cases that the colloidal size mismatches the radius of curvature of the valleys on the wall. Previously the authors published two papers focused on the case when these sizes match. The results in this manuscript are generally important towards realizing the goal that by designing a director field, one can in principle engineer a landscape of elastic energy for the embedded colloidal particles and directing their assembly and sorting, and thus are of great interest to a large community in self-assembly, colloidal science, liquid crystals. Therefore, I think this manuscript is publishable in Nature Communications in terms of its scientific contents. However, the manuscript seems to be not well organized, and I have a hard time to follow through the whole text. I would recommend the authors to consider a major revision in terms of presentations of the results.

(1) The description of the wall structure should be included in the main text, instead of in the supplementary materials, i.e. Fig. S1a should be in Fig. 1 in the main text. In addition, in the first paragraph of Results, the parameter R and A are not defined in the text, and should be defined before mentioned.

(2) Figures need to be labeled in the sequence they described in the main text. Fig. 1 contains results that are discussed in several subsections. It is hard for the readers to follow. For example, in the section "Quadrupole to dipole transition", one has to go back to Fig. 1c to see the images, and then go to Fig. 4. Flipping the papers back and forth to find these images is a little distracting from enjoying the nice science. Another example, Fig. 1f and 1g are quite in conflict with Fig. 1c in their appearance, the readers need to read way back to figure out what they are meant for.

(3) I did not find the definition of the polar angle θ . It will be nice if the authors can include the definition in some of the schematic drawing such as Fig. 3a.

(4) The simulated particle and system size are one order of magnitude smaller than the experimental system. It would be nice if the authors can explain how the interactions and equilibrium positions scale with the system size, and why they would expect that the energy potentials and He/a should be similar to the experimental systems (as they are compared in Fig. 3).

(5) Several paragraphs are quite long, and contain a number of ideas or themes. It will be nice if each paragraph can have one major theme.

Reviewer #2 (Remarks to the Author):

The manuscript presents detailed experimental and numerical studies on colloidal motion in confined nematic liquid crystal systems. The elastic energy fields set up in nematic liquid crystals near wavy walls are used to manipulate the range of attraction, define stable, metastable, and unstable equilibria, and drive transitions between topological defect configurations; the numerical analysis provides valuable insights into the physics behind the observed behavior. Overall, the paper is well written and the subject matter engaging.

However, more data (or more analysis of existing data) on the preferred docking sites of particles in a weak flow along the wavy wall, i.e. the "Goldilocks" wells, would be very much appreciated. Currently, this portion of the manuscript is perhaps the most intriguing but, unfortunately, the least well developed. Similarly, a more developed discussion of the colloidal motion with reference to an electrostatic analogy would likely be extremely instructive, e.g. how much of the behavior can be understood as that of electrostatic dipoles and quadrupoles within non-uniform electrostatic fields, and if and when this analogy break downs.

Notwithstanding these two points, the presented approach is novel and the manuscript is

recommended for publication in Nature Communications, provided the following points have been addressed:

1. The parameter λ is referred to as both the “period” and “wavelength” of the wavy wall interchangeably. It would help the reader if a term were used. Perhaps “period” would be the more appropriate.
2. In the caption for Figure 1 (a), it is stated that the “reference energy [is] evaluated at $(\lambda/2, \lambda)$ ”. However, the term λ has yet to be defined.
3. In the caption for Figure 1 (b), it would be useful if the parameters of the wall, i.e. R , A , and λ , where also named (e.g. radius of curvature, amplitude, and period).
4. In Figure 3, it would help the reader if sub-figures (a) and (e) were labelled with the defect types to which they refer (e.g. Saturn ring, distorted Saturn ring, and hedgehog), and if a legend for the data markers were displayed within the sub-figures (e.g. center of mass, HCOM, and defect position, Hdefect).
5. In the caption for Figure 3, and elsewhere, it would be useful if multiple terms for the same phenomena were minimized, e.g. “distorted Saturn rings”, “displaced defect case”, and “defect displacement cases”.
6. In Figure 4, it would be useful if all angular data were presented in degrees or radians, but not a mixture of the two. Similarly, it would be useful if the axes for angular data in degrees highlighted both 90° and 180° if appropriate, e.g. Figure 4 (a). Finally, it may help the reader if that data which is reproduced from Ref. [41] was more clearly labelled as such within the figure, and if it occupied a single row or column.
7. In Figure 5 and the associated text, it would be useful if the authors could motivate the calculation of the energy density for dipole- and quadrupole-type defects at two different heights, i.e. at $y = 1.5a$ and $1.8a$, respectively. The relationship between the four sub-figures (a)—(d) and sub-figure (e) is also not as clear as it could be. It is currently too easy for the reader to erroneously interpret sub-figure (e) as a plot of absolute energy density for three of the four cases presented in sub-figures (a)—(d), rather than a plot of the difference in energy density between the dipole- and quadrupole-type defects. Improved labelling of the sub-figure should help this, e.g. by including the definition of ΔE in the figure itself. Finally, is there a particular reason to present the data in sub-figure (e) as a function of colloidal volume rather than, say, radius?
8. In Figure 8, it would be helpful to indicate the direction and (average) rate of the flow.

Reviewer #3 (Remarks to the Author):

This paper reports on the approach for controlling the migration of colloidal particles by designing energy landscapes in confined nematic liquid crystals via undulated surfaces. The undulated - wavy- surfaces create spatially variable distortions of nematic orientational order, which in turn creates effective energy landscape for the colloidal particles. Methodologically, the work is a combination of experiments and numerical modelling, with the manipulation of the particles achieved by using magnetic particles and external magnetic fields. Specifically, the authors show controllable trapping of colloidal particles by surface undulations with tunable equilibrium position, undulation affected transition between (dipolar and quadrupolar) particle configurations, and flow stimulated docking of particles into preferred surface wells. In general, the paper is written to good degree and the analysis is clearly performed. Overall, I am rather positive about this work; however, I believe there are some queries which the authors should consider:

- 1) As my main major comment about the paper, I think the idea pushed forward by the paper –to

use structured surfaces for colloidal assembly-is good and interesting. However, what I miss in the paper is some of the wow effect that is usually associated with Nature family publications. For example, the authors use a notable part of the paper to discuss about elastic dipole to quadrupole transition, which is a topic that was looked from various angles quite extensively already in the past. On the other hand, the dynamic docking (authors call it Goldilocks) is brushed through very quickly, although to my opinion new and exciting. Also, the authors –I am rather sure with their state of the art control of microfluidic channels- could perhaps think of pushing somewhat further with the complexity of the undulations of the channels, i.e. beyond rather simple wavy. For example, what if undulations in a distinct channel are composed of multiple wavelengths, or what if some undulations are sharp, with the extreme regime being fractal surfaces in contact with nematic. I understand that many of these ideas go beyond the paper, but I think any addition in this direction would add to the impact of the paper.

2) In addition to geometry, surface anchoring and any nematic elastic anisotropy are major parameters that control such energy landscapes. In the paper, I would suggest to add some more elaborate discussion on their role.

This manuscript was reviewed by three referees, who were generally positive about this submission. Each referee, however, gave detailed comments and suggestions to improve the work. We thank them for these comments, which are addressed in detail below.

Reviewer #1 (Remarks to the Author):

This manuscript presents a number interesting observations in terms of interactions between colloidal particles and wavy walls, in the cases that the colloidal size mismatches the radius of curvature of the valleys on the wall. Previously the authors published two papers focused on the case when these sizes match. The results in this manuscript are generally important towards realizing the goal that by designing a director field, one can in principle engineer a landscape of elastic energy for the embedded colloidal particles and directing their assembly and sorting, and thus are of great interest to a large community in self-assembly, colloidal science, liquid crystals. Therefore, I think this manuscript is publishable in Nature Communications in terms of its scientific contents. However, the manuscript seems to be not well organized, and I have a hard time to follow through the whole text. I would recommend the authors to consider a major revision in terms of presentations of the results.

We have restructured the paper following this helpful advice. Specifically, we have eliminated the former Figure 1 and replaced with a representation of our experimental system. We have also shortened paragraphs throughout the text to improve the readability of the paper, e.g. the discussion of quadrupole to dipole transition (page 5-7).

(1) The description of the wall structure should be included in the main text, instead of in the supplementary materials, i.e. Fig. S1a should be in Fig. 1 in the main text. In addition, in the first paragraph of Results, the parameter R and A are not defined in the text, and should be defined before mentioned.

The old Fig. S1 has been moved to the main text (page 2). R , A and T are now defined in Fig. 1 caption as well as the main text (page 2-3).

(2) Figures need to be labeled in the sequence they described in the main text. Fig. 1 contains results that are discussed in several subsections. It is hard for the readers to follow. For example, in the section "Quadrupole to dipole transition", one has to go back to Fig. 1c to see the images, and then go to Fig. 4. Flipping the papers back and forth to find these images is a little distracting from enjoying the nice science. Another example, Fig. 1f and 1g are quite in conflict with Fig. 1c in their appearance, the readers need to read way back to figure out what they are meant for.

The paper has been restructured. The subfigures in old Fig. 1 has been removed. The elements of this figure have been placed in the section most relevant to them. These changes are reflected throughout the revised text.

(3) I did not find the definition of the polar angle θ . It will be nice if the authors can include the definition in some of the schematic drawing such as Fig. 3a.

We added a schematic in Fig. 4 to graphically shown the definition of polar angle θ , Fig. 4d (page 5).

(4) The simulated particle and system size are one order of magnitude smaller than the experimental system. It would be nice if the authors can explain how the interactions and equilibrium positions scale with the system size, and why they would expect that the energy potentials and He/a should be similar to the experimental systems (as they are compared in Fig. 3).

We thank the reviewer for the question. The simulation of the colloid with the distorted Saturn ring has been removed from the figures reporting the equilibrium height of the colloid. We have made no direct comparison of the Saturn ring height and simulation. This has been clarified in the text.

In addition, we have addressed the manner in which the simulation scales with colloid radius to clarify where we can and cannot compare it to experiment. Our experiments are too large to be accurately reproduced in simulations, and must be re-scaled with care owing to the owing to the correlation length, which does not scale with system size. This limits our direct simulations to length scales at which the dipole is more energetically costly than the Saturn ring configuration. Thus, we cannot directly simulate the transition from Saturn to dipole configurations.

We have explored how the energy of the system scales with the size of the colloid. That discussion is presented on pages 7 in the revised document, and in Figures 5f and Figure S6 in Supplementary Material, reproduced below. We briefly recapitulate key points here. We calculate ΔE , the difference between the energy of the dipole and the energy of the quadrupole for several colloid radii. The total energy in the system consists of two parts, the phase free energy, i.e., the defect energy, and the gradient free energy, analogous to the distortion of the field. The hedgehog defect does not grow with the system size, while the Saturn ring grows with the linear dimension of the system. Thus, the difference in the phase free energy ΔE_{phase} between dipole and quadrupole is always linear in colloid radius a (Fig. S6a). The difference in the gradient free energy $\Delta E_{gradient}$ is expected to be a sum of a linear term and a linear-times-logarithm term, based on previous theoretical work in simpler geometries (Stark, Phys. Rep. 2001); $\Delta E_{gradient}$ vs colloid radius is fitted to such a form: $k_1 a + k_2 a \log a + k_3$ (Fig. S6b). In the limit of large particle radius, comparable to those in experiment, this fitted form is also linear in a .

We calculate ΔE versus a (Figure 5f) and extrapolate, using the linear form of ΔE_{phase} and the log-linear form of $\Delta E_{gradient}$ fitted to the numerical results. For particles similar to those in experiment, this yields a linear scaling in a . This extrapolation suggests that at large a , the dipole becomes the lower energy state, and is more favored near the wall, in keeping with experiment.

Supplementary Figure S6. Simulation scaling. The simulation of ΔE , the difference between the energy of the dipole and the energy of the quadrupole, has two contributions. These include the phase free energy difference ΔE_{phase} associated with the defects, and the gradient free energy difference $\Delta E_{gradient}$ associated with differences in the distortion of the director field. (a) ΔE_{phase} scales linearly in a . This scaling emerges

because the defect energy of the dipole does not grow while that of the Saturn ring defect does grow with the linear dimension of the system. (b) $\Delta E_{\text{gradient}}$ has a linear part ($\sim a$) and a logarithmic part ($\sim a \log a$) based on [Stark, Phys. Rep., 2001]. Here it is fitted to form $k_1 a + k_2 a \log a + k_3$. At large radius, this form is linear in a .

Figure 5f. LdG simulation of the energy density for dipole and quadrupole near a wavy boundary. ΔE , the difference in energy of the dipole and quadrupole, is calculated for systems of different size (colloid radius $a = 90, 135, 180, 225, 270$ nm, the simulation box and the walls are scaled accordingly) for different distances from the wall y/a . Circles denote simulation results, solid lines are fitted to forms suggested by scaling arguments, dotted line are extrapolations based on these fits.

(5) Several paragraphs are quite long, and contain a number of ideas or themes. It will be nice if each paragraph can have one major theme.

We have made this change to improve the manuscript. For example, in the discussion of equilibrium position, we split up discussion into large and small wells (page 4). In the quadrupole-to-dipole transition section, we split up the discussion into experimental and simulations (page 5-7).

Reviewer #2 (Remarks to the Author):

(1.) The manuscript presents detailed experimental and numerical studies on colloidal motion in confined nematic liquid crystal systems. The elastic energy fields set up in nematic liquid crystals near wavy walls are used to manipulate the range of attraction, define stable, metastable, and unstable equilibria, and drive transitions between topological defect configurations; the numerical analysis provides valuable insights into the physics behind the observed behavior. Overall, the paper is well written and the subject matter engaging.

However, more data (or more analysis of existing data) on the preferred docking sites of particles in a weak flow along the wavy wall, i.e. the “Goldilocks” wells, would be very much appreciated. Currently, this portion of the manuscript is perhaps the most intriguing but, unfortunately, the least well developed.

We thank the reviewer for the suggestion to explore the “Goldilocks” phenomenon in greater detail. We have significantly improved our understanding of this phenomenon by introducing an external potential gradient to drive the colloid, and by studying the competition between this external field and the nematic

director field. We did this by tilting the stage carefully, allowing the colloid to move under gravity parallel to the wall, and to interact in a more complex manner with the director field. The new discussion has been incorporated from page 9-11 in our paper.

We recapitulate key aspects here.

Placing wells of different sizes adjacent to each other offers additional opportunities for path planning. In one setting, a colloid can sample multiple wells of varying sizes under a background flow in the x direction. We followed a colloid moving under the effect of gravity. The sample was mounted on a custom- made holder that can be tilted by an angle α (Fig. 8a, b) within a range between 10° and 20° so that the colloid experiences a body force in the x-direction. We have verified in independent experiments that, without the wall, the particle moves at a constant velocity due to balance of drag and gravity. In the presence of the wavy wall, the particle's trajectory is influenced by the energy landscape there. We first describe the particle paths over a series of periodic wells, and then describe motion for wells of decreasing wavelength.

Docking or continued motion in the cell is determined by a balance between the body force and viscous forces that drive x-directed motion, the range and magnitude of attractive and repulsive elastic interactions with the wall, and viscous drag near the wall. If the particle moves past the well in the x-direction faster than it can move toward the wall, it will fail to dock. However, if attraction to the well is sufficiently pronounced to move the particle there before it flows past, the particle will dock.

For a tilted sample with a wavy wall of uniform wavelength ($\lambda = 70 \mu\text{m}$), colloids initially close enough to the wall dock into the nearest well (Fig. 8c, $\alpha = 11.2^\circ$, Supplemental Video V9). Far from the wall, the particle does not dock. However, the influence of the wall is evident by the fact that the particle does not remain at a fixed distance from the wall. Rather, the distance from the wall varies periodically, and this periodic motion has the same wavelength as the wall itself (Fig. 8d, $\alpha = 12.8^\circ$, Supplemental Video V10).

To simulate the forces on the particle, particles are placed at different locations near a wall, and the energy of the system is calculated (as detailed in Supplemental Material). Gradients in this energy capture the forces on the colloid owing to the liquid crystal at each location. A uniform body force in the x-direction is then added on the colloid to find the trajectories. We simulated the trajectories for various initial loci. We find two outcomes: for strong x-directed force and/or far from the wall, the particle follows a wavy path; for weak -x-directed force and near the wall, the particle docks (Fig. 8c). A particle slows before the hill and moves faster as it approaches the next well. This velocity modulation can be attributed to the interaction with the splay-bend region, similar to particles moving within an array of pillars [Chen et al., *Soft Matter*, 2018]. Our experiments and simulated trajectories are in good agreement, showing the two behaviors. Particles either follow wavy paths in the x direction, or dock in the nearest well. (Fig. 8e).

However, a different behavior was observed when we modulate the wavelength of the wavy wall. As a particle travels past successive wells of decreasing periods ($\lambda = 70, 60, 50, 40 \mu\text{m}$, Supplemental Video V11), the particle moves in the y direction, closer to the wells, until it eventually is entrained by a steep enough attraction that it docks (Fig. 8f, $\alpha = 12.3^\circ$). This particle, like Goldilocks, protagonist of a beloved children story, finds the well that is “just right”. Simulation of two wells with different wavelengths and a superimposed force confirms these results: we can achieve an additional state not possible with the uniform well, i.e. a wavy trajectory that descends and docks (Fig. 8e, yellow curve). The slight energy difference between wells of different wavelength underlies the “Goldilocks” phenomenon. Since the energy landscape exploits bend and splay, the ratio between the elastic constants K_{11} and K_{33} is important in determining the particle paths.

Figure 8: “Goldilocks”: A colloid in motion docks in a preferred well. (a) Schematic of the experimental setup with tilt angle α to harness gravity to drive colloid motion in x-direction. (b) Image of the setup which allows fine control of the tilt, and thus the x-component of gravitation constant $g \sin \alpha$ (c) A colloid traveling at $\bar{V}_x = 0.01 \mu\text{m/s}$ at tilt angle $\alpha = 11.2^\circ$. (d) A colloid traveling at $\bar{V}_x = 0.06 \mu\text{m/s}$ at tilt angle $\alpha = 12.3^\circ$. (e) Particle trajectories predicted for the sum of the elastic energy field and a gravitational potential energy gradient in -x-direction across uniform set of wells ($\lambda = 70 \mu\text{m}$). (f) A colloid traveling at $\bar{V}_x = 0.09 \mu\text{m/s}$ at tilt angle $\alpha = 12.8^\circ$ across wells of decreasing wavelengths ($\lambda = 70, 60, 50, 40 \mu\text{m}$). (g) Particle trajectories predicted for the sum of the elastic energy field and a gravitational potential energy gradient in -x-direction). All colloids have $2a = 13 \mu\text{m}$. The scale bars are $20 \mu\text{m}$.

(2) Similarly, a more developed discussion of the colloidal motion with reference to an electrostatic analogy would likely be extremely instructive, e.g. how much of the behavior can be understood as that of electrostatic dipoles and quadrupoles within non-uniform electrostatic fields, and if and when this analogy break downs.

Our experimental system would indeed provide an ideal platform to test the limitations of the electrostatic analogy. One of the key factors for this is that our imposed field does not contain any topological defects. We could, for example, use the method by Chernyshuk and Lev [Phys. Rev. E. 84, 011707 (2011)] to calculate the interaction between the colloids and the wall. However, the geometry of the wall is non-trivial and the Green’s function method is beyond the scope of this paper.

The analogy with electrostatic multipoles is expected to break down in the vicinity of the wall and the particle where gradients in the director become steep, which is the zone of interest in this research. For this reason, we do not use an approximate analysis based on multipoles to analyze our results. Rather, we simulate the three-dimensional director field in a Q tensor (Landau de Gennes) formulation to find equilibrium states for the particle in different locations in the domain. Since the colloids move at negligible Reynolds and Erickson numbers, their motion is quasi-equilibrium; spatial gradients in this equilibrium energy field determine the forces on the particles.

(3.) Notwithstanding these two points, the presented approach is novel and the manuscript is recommended for publication in Nature Communications, provided the following points have been addressed:

3.1. The parameter λ is referred to as both the “period” and “wavelength” of the wavy wall interchangeably. It would help the reader if a term were used. Perhaps “period” would be the more appropriate.

We appreciate the comment. We have adopted the term “wavelength” for the length scale λ throughout the text. We have chosen to use "wavelength" over "period" because this quantity was often used in comparison with other length-scales, therefore the use of "wavelength" was more immediate.

3.2. In the caption for Figure 1 (a), it is stated that the “reference energy [is] evaluated at $(\lambda/2, \lambda)$ ”. However, the term λ has yet to be defined.

λ is now defined first when we describe system geometry (page 2), and again in the Multiple states section (page 8).

3.3. In the caption for Figure 1 (b), it would be useful if the parameters of the wall, i.e. R, A, and λ , were also named (e.g. radius of curvature, amplitude, and period).

The parameters are now named in the caption Fig. 1a (page 2).

(4.) In Figure 3, it would help the reader if sub-figures (a) and (e) were labelled with the defect types to which they refer (e.g. Saturn ring, distorted Saturn ring, and hedgehog), and if a legend for the data markers were displayed within the sub-figures (e.g. center of mass, HCOM, and defect position, Hdefect).

Fig 3 title and legends have been added. Further, for clarification, we now present experiment data and schematics side-by-side. The definition of y is first presented on page 2, then again on Fig. 3 and text associated with the equilibrium position section (page 4). In addition, the nomenclature for the position of the colloid and defects are now presented in terms of “ y ” with an appropriate subscript.

(5.) In the caption for Figure 3, and elsewhere, it would be useful if multiple terms for the same phenomena were minimized, e.g. “distorted Saturn rings”, “displaced defect case”, and “defect displacement cases”.

We now refer to “distorted Saturn rings” (Fig. 3, page 4). We have also added schematics to make the definition of the “distorted Saturn ring” clearer.

(6.) In Figure 4, it would be useful if all angular data were presented in degrees or radians, but not a mixture of the two. Similarly, it would be useful if the axes for angular data in degrees highlighted both 90° and 180° if appropriate, e.g. Figure 4 (a). Finally, it may help the reader if that data which is reproduced from Ref. [41] was more clearly labelled as such within the figure, and if it occupied a single row or column.

In Fig. 4 (page 5), we have relabeled the axis so the angles are all expressed in degrees, and changed the axis so that it now only contains 90° (Saturn ring) and 180° (dipole) for simplicity. (3) We apologize for the confusion, in fact, the data from Ref. [41] was not reproduced in our paper. However, our results bear striking resemblance.

(7.) In Figure 5 and the associated text, it would be useful if the authors could motivate the calculation of the energy density for dipole- and quadrupole-type defects at two different heights, i.e. at $y = 1.5a$ and $1.8a$,

respectively. The relationship between the four sub-figures (a)—(d) and sub-figure (e) is also not as clear as it could be. It is currently too easy for the reader to erroneously interpret sub-figure (e) as a plot of absolute energy density for three of the four cases presented in sub-figures (a)—(d), rather than a plot of the difference in energy density between the dipole- and quadrupole-type defects. Improved labelling of the sub-figure should help this, e.g. by including the definition of ΔE in the figure itself. Finally, is there a particular reason to present the data in sub-figure (e) as a function of colloidal volume rather than, say, radius?

We select the two heights, $y = 1.5a$ and $y = 1.8a$ because they are determined to be the equilibrium heights for dipole and Saturn ring, respectively, for constants in the LdG simulation corresponding to high core energy for which the dipole can be simulated.

The definition of ΔE has now been included in Fig. 5 itself (page 6).

The difference of the total free energy (ΔE) is plotted as a family of curves for different positions y versus the radius of the colloid Fig. 5 (page 6). The scaling of the contributions to this quantity are discussed in detail. The dashed lines in this figure are extrapolations according to that scaling; this extrapolation suggests that, at large radius, the dipole becomes the stable state.

A detailed discussion of the dependence of the simulation on particle radius is provided in the text and in our response to Referee 1, point 4, above.

Figure 5f. LdG simulation of the energy density for dipole and quadrupole near a wavy boundary. ΔE , the difference in energy of the dipole and quadrupole, is calculated for systems of different size ($a = 90, 135, 180, 225, 270$ nm, the simulation box and the walls are scaled accordingly) for different distances from the wall y/a . Scaling arguments guide extrapolation to large particle radius.

(8.) In Figure 8, it would be helpful to indicate the direction and (average) rate of the flow.

We now impose a body force on the particle by tilting the stage and exploiting gravity. The average velocity of the particle in the x-direction \bar{v}_x is indicated in the caption in Fig. 8 (page 10).

Reviewer #3 (Remarks to the Author):

This paper reports on the approach for controlling the migration of colloidal particles by designing energy landscapes in confined nematic liquid crystals via undulated surfaces. The undulated -wavy- surfaces create spatially variable distortions of nematic orientational order, which in turn creates effective energy landscape for the colloidal particles. Methodologically, the work is a combination of experiments and numerical modelling, with the manipulation of the particles achieved by using magnetic particles and external magnetic fields. Specifically, the authors show controllable trapping of colloidal particles by surface undulations with tunable equilibrium position, undulation affected transition between (dipolar and quadrupolar) particle configurations, and flow stimulated docking of particles into preferred surface wells. In general, the paper is written to good degree and the analysis is clearly performed. Overall, I am rather positive about this work; however, I believe there are some queries which the authors should consider:

1) As my main major comment about the paper, I think the idea pushed forward by the paper –to use structured surfaces for colloidal assembly-is good and interesting. However, what I miss in the paper is some of the wow effect that is usually associated with Nature family publications. For example, the authors use a notable part of the paper to discuss about elastic dipole to quadrupole transition, which is a topic that was looked from various angles quite extensively already in the past. On the other hand, the dynamic docking (authors call it Goldilocks) is brushed through very quickly, although to my opinion new and exciting. Also, the authors –I am rather sure with their state of the art control of microfluidic channels- could perhaps think of pushing somewhat further with the complexity of the undulations of the channels, i.e. beyond rather simple wavy. For example, what if undulations in a distinct channel are composed of multiple wavelengths, or what if some undulations are sharp, with the extreme regime being fractal surfaces in contact with nematic. I understand that many of these ideas go beyond the paper, but I think any addition in this direction would add to the impact of the paper.

We thank the reviewer for this comment. Indeed, the research directions indicated by the reviewer are very interesting and some of that work is currently in progress (e.g. sharp undulations). We prefer not to include the discussion of sharp undulations in this paper because we intend to limit ourselves to a non-singular director field here, where defects associated with the wall do not play a major role.

Guided by these comments, we have further developed the “Goldilocks” concept by showing the competition between an external field (gravity) and the elastic energy field at the wavy wall.

The new discussion has been incorporated from page 9-11 in our paper. We give key points in our response to Referee 2, point 1, above.

Figure 8: “Goldilocks”: A colloid in motion docks in a preferred well. (a) Schematic of the experimental setup with tilt angle α to harness gravity to drive colloid motion in x-direction. (b) Image of the setup which allows fine control of the tilt, and thus the x-component of gravitation constant $g \sin \alpha$ (c) A colloid traveling at $\bar{V}_x = 0.01 \mu\text{m/s}$ at tilt angle $\alpha = 11.2^\circ$. (d) A colloid traveling at $\bar{V}_x = 0.06 \mu\text{m/s}$ at tilt angle $\alpha = 12.3^\circ$. (e) Particle trajectories predicted for the sum of the elastic energy field and a gravitational potential energy gradient in -x-direction across uniform set of wells ($\lambda = 70\mu\text{m}$). (f) A colloid traveling at $\bar{V}_x = 0.09 \mu\text{m/s}$ at tilt angle $\alpha = 12.8^\circ$ across wells of decreasing wavelengths ($\lambda = 70, 60, 50, 40 \mu\text{m}$). (g) Particle trajectories predicted for the sum of the elastic energy field and a gravitational potential energy gradient in -x-direction). All colloids have $2a = 13 \mu\text{m}$. The scale bars are $20 \mu\text{m}$.

2) In addition to geometry, surface anchoring and any nematic elastic anisotropy are major parameters that control such energy landscapes. In the paper, I would suggest to add some more elaborate discussion on their role.

We thank the reviewers for raising the two interesting points concerning (1.) anchoring strength and (2.) elastic constants. These are indeed important parameters we can vary if we decide to use different surface treatment protocols or different liquid crystals.

Supplementary Figure S9. Effect of anchoring. The energy of a colloid with Saturn ring defect is simulated by placing it at different distances above a well in a cell of $T = 4a$. For realistic anchoring strength, the energy profile as the particle position changes near the wall remains very similar to the case of infinite anchoring (solid line). If we decrease the anchoring by 10-fold, binding energy (the energy difference between when the particle is far and when it is near the well) decreases, so does the gradient. Therefore, we also expect the range of interaction to decrease.

(1.) The extrapolation length of our system is $\xi = K/W \approx 1 \mu\text{m}$, which is small compare to the experimental scales (10s of μm), meaning we are in the large W regime. Therefore, we can take W to be infinite when scaling down the system size in the numerics.

To explore the role of finite anchoring energies, LdG simulation is used to calculate the total energy of the system with a colloid with homeotropic anchoring in a Saturn ring configuration. The colloid of radius a is placed at various distances y from the wavy wall, centered above the well, and the anchoring strength is systematically varied. Under realistic anchoring strengths ($10^{-3} - 10^{-2} \text{J/m}^2$), the behavior of the energy of moving a colloid from near to far does not deviate much from that in the case of infinite anchoring (solid black line). However, as we further decrease the anchoring, the binding energy decreases, as does the range of the interaction between the particle and the well (Fig. S9). A discussion of the effect of anchoring is added in Supplemental Material and main text on page 13.

(2.) We used the one elastic constant approximation to run all of our simulations. Since we have a quasi-2D system, with the director field expected to lie in the plane of the wavy wall, the effect of changing the twist constant is expected to be weak in comparison to changing the splay and bend elastic constants. Changing anchoring and elastic constant are expected to change much of the behavior of the system.

We expect that a system with $K_{11} > K_{33}$ might weaken the docking interactions and will alter the range of interaction above the wells, since splay will cost more energy. Furthermore, since horizontal migrations force the particle to interact with domains of strong bend above the inflection points on the wall, we expect a system with $K_{11} > K_{33}$ will penalize splay and has increased repulsion from these regions. This would therefore influence particle trajectories and the ‘‘Goldilocks’’ experiments. Conversely, $K_{11} < K_{33}$ is expected to strengthen the docking interactions.

The ratio between K_{11} and K_{33} is also crucial in determining the equilibrium position of the particle and the transition between dipole and quadrupole. As the bend constant increases, the particle is expected to equilibrate closer to the wall to be in splay matching. The bend distortion is also responsible for distorting and displacing the Saturn ring, eventually leading to the transition to dipole.

The importance of this ratio is now discussed in the text in several places, specifically:

- The equilibrium positions, page 5
- The critical angle for quadrupole to dipole transition, which can be reason why we measure a different angle from [Škarabot *et al.*, Phys. Rev. E, 2008, 031705], page 6
- The effect on “Goldilocks”, or site selective docking, page 11
- A discussion of simulation parameters in the Method section on page 13.

REVIEWERS' COMMENTS:

Reviewer #1 (Remarks to the Author):

The authors have made major revisions and have addressed all the issues raised. I recommend it for publication.

Reviewer #2 (Remarks to the Author):

The authors have clearly gone to considerable lengths to address all reviewers' comments, resulting in a much-improved manuscript. Having fully satisfied any previous concerns, it is therefore recommended that the manuscript be published in Nature Communications without further revision.

Reviewer #3 (Remarks to the Author):

The authors have very well and extensively addresses my comments. I recommend publication.